# Microarray Profiling of Differentially Expressed Genes in Coronary Artery Bypass Grafts of High-Risk Patients with Postoperative Cognitive Dysfunctions

**DOI:** 10.3390/ijerph20021457

**Published:** 2023-01-13

**Authors:** Noor Anisah Abu Yazit, Norsham Juliana, Suhaini Kadiman, Kamilah Muhammad Hafidz, Nur Islami Mohd Fahmi Teng, Nazefah Abdul Hamid, Nadia Effendy, Sahar Azmani, Izuddin Fahmy Abu, Nur Adilah Shuhada Abd Aziz, Srijit Das

**Affiliations:** 1Faculty Medicine and Health Sciences, Universiti Sains Islam Malaysia, Nilai 71800, Malaysia; 2Anaesthesia and Intensive Care Unit, National Heart Institute, Kuala Lumpur 50400, Malaysia; 3Faculty of Health Sciences, Universiti Teknologi MARA, Puncak Alam 42300, Malaysia; 4Institute of Medical Science Technology, Universiti Kuala Lumpur, Kajang 43000, Malaysia; 5Department of Human & Clinical Anatomy, College of Medicine & Health Sciences, Sultan Qaboos University, Al-Khoud, Muscat 123, Oman

**Keywords:** cognitive decline, microarray, differentially expressed genes, CABG

## Abstract

Postoperative cognitive dysfunction (POCD) is cognitive decline after surgery. The authors hypothesized that gene-level changes could be involved in the pathogenesis of POCD. The present study evaluated the incidence of POCD and its associated differentially expressed genes. This was a prospective cohort study conducted on high-risk coronary artery bypass graft patients aged 40 to 75 years. POCD classification was based on a one standard deviation decline in the postoperative scores compared to the preoperative scores. The differentially expressed genes were identified using microarray analysis and validated using quantitative RT-PCR. Forty-six patients were recruited and completed the study. The incidence of POCD was identified using a set of neurocognitive assessments and found to be at 17% in these high-risk CABG patients. Six samples were selected for the gene expression analyses (3 non-POCD and 3 POCD samples). The findings showed five differentially expressed genes in the POCD group compared to the non-POCD group. The upregulated gene was ERFE, whereas the downregulated genes were KIR2DS2, KIR2DS3, KIR3DL2, and LIM2. According to the results, the gene expression profiles of POCD can be used to find potential proteins for POCD diagnostic and predictive biomarkers. Understanding the molecular mechanism of POCD development will further lead to early detection and intervention to reduce the severity of POCD, and hence, reduce the mortality and morbidity rate due to the condition.

## 1. Introduction

Postoperative cognitive dysfunction (POCD) is a mental impairment that arises after a surgical procedure. It is defined by a range of changes in neurocognitive patients’ condition and behavior for weeks or even months from the time anesthesia is introduced in the intensive care unit until a few months after the surgery [1]. Comparing types of surgery, the prevalence of POCD in cardiac surgery is much higher than in non-cardiac surgery. A study reported that cardiac surgery has a 40% prevalence of POCD compared to 20% for non-cardiac surgery [2]. A meta-analysis discovered that adjuncts in anesthesia, such as dexmedetomidine, may alleviate or reduce the prevalence of POCD in cardiac surgery [3].

Patients are diagnosed with POCD when there are post-operative deficits in their mental states, such as attention, concentration, executive function, memory, visuospatial ability, and psychomotor speed. POCD brings disadvantages to patients’ postoperative care, such as prolonged rehabilitation and hospitalization, reduced quality of life, increased functional recovery time, and increased healthcare costs [4]. The diagnosis requires sensitive and tedious pre-operative and post-operative neuropsychiatric tests. Up to date, multiple standard cognitive tests were utilized to indicate the problem, however, there are no gold standard tests being exclusively set only for POCD. The complication may result in an increased risk of losing their jobs or their independence and leads to a significant reduction in their quality of life [5]. However, the exact mechanism of POCD is unclear due to conflicting results and a lack of evidence [6]. Despite the fact that POCD was discovered a long time ago, there is an ongoing debate regarding the risk factors, diagnostic criteria, pathophysiology, outcomes, and various effects on the effective postoperative management of patients.

The study of POCD continues as researchers actively study gene changes in POCD. Changes in gene levels caused by surgery and anesthesia, according to the authors, play a critical role in the pathogenesis of POCD [7,8]. Often, the diagnosis of POCD is difficult and tedious, with no gold standard methods, classification, or molecular diagnosis [9]. This opens up the opportunity for researchers to explore its gene expression profiling, identification of target genes, construction of interaction network, and pathway analysis. Studies focusing on gene changes with regard to POCD were still lacking, thus allowing authors to explore the area.

The urge to find how POCD occurs grows, as the mechanisms of POCD development are still vague as they are related to various factors. The focus among researchers in the past few years has been neuroinflammation and POCD. Li et al. (2022) describe the pathways by which peripheral inflammation leads to neuroinflammation. Inflammation and immune activation due to surgical stress will release the damage-associated molecular patterns (DAMPs), which include high mobility group protein B1 (HMGB1). HMGB1 secretions were enhanced by proinflammatory factors that induce and maintain the peripheral inflammatory response [10]. The peripheral inflammation may spread to the central nervous system via a specific transporter, periventricular area, blood–brain barrier disruption, homoreceptor, and vagal nerve [6], thereby causing neuroinflammation and cognitive dysfunction. The finding was in accordance with a study where HMGB1 levels in serum were reported to be higher in POCD patients compared to the control group, showing the potential correlation between HMGB1 and POCD [11].

With current technological advancements, many medical breakthroughs have been discovered, especially in understanding disease pathophysiology at a molecular level. Gene studies have been involved in disease diagnosis, prognosis, and risk factor assessment [12]. Apolipoprotein Epsilon-4 (APOE4), one of the genes closely associated with Alzheimer’s disease, was thoroughly studied to determine its relationship with POCD. However, the results were conflicting and had a poor prognosis [13]. Hence, current studies focus on differentially expressed genes to understand the mechanism involved in POCD. A study has summarized different types of microribonucleic acid (miRNA) expressions involved in POCD mechanisms, mainly through neuroinflammation [7]. Another study also found that long non-coding RNA (lncRNA) and messenger RNA (mRNA) were differentially expressed, suggesting their involvement in POCD pathogenesis [9]. In addition, Wang et al. (2019) identified an abnormal expression of circular RNA (circRNA_089763) that was found in POCD patients using the microarray technique [14].

The present study aimed to identify the incidence of POCD using specific neurocognitive assessments and to understand its associated key differentially expressed genes using the microarray technique, in high-risk coronary artery bypass graft patients that received dexmedetomidine in the perioperative period. The findings from this study are novel as understanding the molecular mechanism of POCD development will further lead to early detection and intervention to reduce the severity of POCD, hence, reducing the mortality and morbidity rate due to the condition.

## 2. Materials and Methods

### 2.1. Study Design and Patients Selection

This prospective cohort study on high-risk CABG patients was conducted in order to elucidate the key differences in gene expressions for the incidence of POCD. The sampling frame includes CABG patients aged 40 to 75 years. We included patients that were categorized as high risk according to the doctor in charge, able to understand Malay or English language, and not diagnosed as having dementia. The exclusion criteria included; Mini-Mental State Examination (MMSE) score < 24 points; patients with planned off-pump surgery; dialysis-dependent patients; patients with severe hepatic impairment; and pregnant women. Written informed consent was obtained from patients prior to the recruitment. This study was conducted with prior ethical approval by the National Heart Institute Ethical Committee (IJNREC/441/2019).

### 2.2. Neurocognitive Assessment

Patients were screened for eligibility using the Malay version of the Mini-Mental State Examination (MMSE) [15]. Only patients with MMSE scores of more than 24 were recruited to eliminate patients with poor cognitive performance at baseline. The recruited patients completed the neurocognitive assessment one day before surgery and seven days after. Patients were admitted 2 days prior to the surgery to allow routine preoperative assessments (blood test, echocardiogram, physiotherapy) to be carried out. The patients were enlightened about all the procedures during their clinic sessions and subsequently, all patients selected in the study were screened to assure they were psychologically stable enough to conduct all the preoperative assessments including the cognitive testing. Furthermore, consent from all patients was taken. Hence, they were free to withdraw from the projects at any point without any discrimination on the standard treatment or patient management protocol. On average, patients were allowed to discharge one week after surgery, or even earlier (depending on the doctor’s assessment); hence, day 7 postoperative is the best time to conduct a cognitive assessment. The neurocognitive assessments utilized in this study were: (1) Trail Making Test (TMT) Part A and Part B for visuoperceptual abilities, secondary task switching, and working memory; (2) Digit Span Forward and Backward to test attention and working memory, (3) Digit Symbol Substitution Test (DSST) to test for speed, attention, and manual dexterity [16]; and (4) Clock Drawing Test (CDT) to assess visuoconstructive abilities. To discriminate on patients’ POCD statuses, the 1 SD method was adopted. The method defines a decline in cognitive function based on more than one standard deviation of the postoperative scores compared to the preoperative scores. The SD was calculated from population norms or published samples. Our population norms were taken from Ibrahim et al. (2009), who validated Malay MMSE in our population, where 1 SD equals 3.6 scores [17].

### 2.3. Data Collection

Demographic data such as age, gender, body mass index (BMI), education level, and intraoperative data such as bypass and cross-clamp times were collected. Blood samples were collected one day before surgery and three days after surgery. Ten ml of blood were collected in anticoagulant tubes and centrifuged at 4 °C at 400× *g* for 5 min. The samples were then subjected to RNA extraction on the same day.

### 2.4. RNA Extraction

The RNA extraction protocol was conducted to extract total RNA from samples. GeneJET^TM^ (Thermo Fisher Scientific Inc., Waltham, MA, USA) Whole Blood RNA Purification Mini Kit was used. Briefly, 50–500 µL of blood was centrifuged at 4 °C at 400× *g* for 5 min, and the supernatant was discarded. The cells were lysed by adding 600 µL of lysis buffer and 450 µL of ethanol using a vortex. The lysate was purified by transferring it to a collection tube inserted with the column, centrifuged at 12,000× *g* for 1 min. Then, the flow-through solution was discarded, and the purification column was transferred to a new collection tube. To eliminate excess protein and impurities, 700 µL of Wash Buffer 1 was added and centrifuged for 1 min at 12,000× *g*. The flow-through solution was discarded, and the process was repeated with 500 µL Wash Buffer 2. Another 500 µL Wash Buffer 2 was added and centrifuged for 2 min at 12,000× *g*. The purification column was then placed into a new 1.5 mL collection tube to collect the RNA. By adding 50 µL of nuclease-free water to the center of the purification column membrane, centrifuge for 1 min at 12,000× *g*. The purified RNA collected in the tube was immediately stored at −80 °C until further analysis.

### 2.5. Sample Selection and Microarray Analysis

Six samples were randomly selected: three non-POCD (labeled A, B, C) and 3 POCD (labeled D, E, F). Microarray analysis was conducted using the Agilent SureScan Microarray Scanner (Agilent Sureprint and SureScan Technologies, Santa Clara, CA, USA). The sample was sent to Neoscience Sdn Bhd Lab (Selangor, Malaysia) for all protocols to be carried out in service provider facilities. The samples were checked for purity and concentration using an Agilent 2100 Bioanalyzer (Agilent Technologies, Santa Clara, CA, USA). The microarray protocol includes labeling the samples, hybridizing, washing, and staining before analysis using fluorescence techniques. The raw data were extracted and analyzed using GeneSpring GX Bioinformatic Software version 13.0. Differentially expressed genes (DEGs) were validated using real-time PCR (qRT-PCR). Our genes of interest were selected for qRT-PCR. The selection of genes of interest was made through a literature search.

### 2.6. Statistical Analysis

Statistical analysis was conducted using SPSS 26.0 software (SPSS, Inc., Chicago, IL, USA). Normality tests were conducted, and normally distributed data were presented as mean ± SD. Data were analyzed using an independent *t*-test for demographic and intraoperative analysis between the POCD and non-POCD groups, and a dependent *t*-test for comparison of before and after surgery. Categorical data were presented in frequency and percentage and analyzed using the Pearson Chi-square test. A *p*-value of <0.05 was considered statistically significant.

## 3. Results

### 3.1. Neurocognitive Assessment

Forty-six patients were recruited and successfully completed the study. Out of 46, 17% (*n* = 8) patients were found to have decreased cognitive assessment scores, hence classified as POCD at discharge. Patients in the POCD group were significantly older and had lower education levels than non-POCD patients (*p* < 0.05). Postoperative cognitive assessment reveals significant differences between the POCD and non-POCD groups, with the POCD group having lower MMSE, digit span, DSST, and CDT scores (*p* > 0.05). All data are shown in Table 1.

The percentage differences between POCD and non-POCD groups for baseline and postoperative scores were calculated for each cognitive test. MMSE, digit span, digit symbol substitution test, and clock drawing test showed that POCD patients had a more significant percentage decrease from baseline compared to non-POCD patients. Regarding the Trail Making Test, an increased percentage difference means the time taken was longer. A longer time taken to complete the Trail Making Test reflects slower visuoperceptual abilities as seen in POCD patients. The graph of the percentage differences for all tests is shown in Figure 1.

### 3.2. Samples Quality Detection

The detection of outliers in our samples was conducted using PCA plots. Our models show a normal distribution and good sample correlations, as illustrated in Figure 2. The purity, concentration, and RNA integrity numbers were tabulated in Table 2.

### 3.3. Differentially Expressed Genes

Microarray analysis showed that five genes were differentially expressed, four were downregulated, and one was upregulated (Table 3). The volcano plot and clustering of the genes are shown in Figure 3. The upregulated gene was ERFE, whereas the downregulated genes include KIR2DS2, LIM2, KIR3DL2, and KIR2DS3. Differential expression was set at fold change > 2 and statistically significant at *p* < 0.05.

### 3.4. Quantitative RT-PCR Validation

mRNA expression levels of three genes (ERFE, KIR2DS3, and KIR3DL2) were analyzed by qRT-PCR. However, two genes were not analyzed; LIM2 was identified as not the gene of interest, whereas KIR2DS2 had insufficient samples to proceed after optimization, respectively, and hence, was discarded from validation.

Microarray analysis revealed that gene expression levels correlated with qRT-PCR analysis, with both showing a similar trend in POCD patients’ genes. Both KIR2DS3 (*p* = 0.021) and KIR3DL2 (*p* = 0.008) genes were found to be significantly different between POCD and non-POCD patients. Gene ERFE (*p* = 0.672), on the other hand, offered a higher expression in POCD patients without ever reaching statistical significance (Figure 4).

## 4. Discussion

Cognitive decline postoperatively is a concern, especially in cardiac surgery. A meta-analysis of 215 studies involving 91,829 patients discovered that 43% of patients experienced cognitive impairment immediately following coronary artery bypass surgery, which decreased to 19% at 4–6 months postoperatively [18]. The meta-analysis involves patients receiving multiple ranges of standard care perioperatively, depending on the clinical site. Compared to our study, the prevalence of POCD at discharge was 17%, and this study homogenously included high-risk patients who received dexmedetomidine sedation.

Yazit et al. (2020) discussed that advancing age would cause degenerative changes, hence becoming one of the risk factors for POCD [13]. Current meta-analyses prove that preoperative depression and old age are significant risk factors for POCD after CABG [19]. Since healthcare advancements improve longevity, many older individuals were able to undergo cardiac surgery. Besides, higher education levels were found to reduce the probability of cognitive decline. Individuals with more exposure to mental challenges and complex tasks were said to have a higher cognitive reserve, lessening the likelihood of POCD [20]. These findings correspond well with our results, whereby patients in the POCD group were significantly older and had lower education levels than non-POCD patients (*p* < 0.05).

Cognitive screening in the perioperative setting presents significant challenges related to time and efficiency. The tools selected for this study are well-validated and reliable neuropsychological measures with excellent sensitivity and specificity for detecting subtle impairment. However, they are still time-consuming and require expertise and experience to administer. Because we recognized the importance of accuracy in cognitive screening and the importance of time management during sampling, the tests chosen for this study supplement the need for cognitive assessment while requiring minimal administration time [21].

A total of five genes were differentially expressed in POCD groups: ERFE, KIR2DS2, LIM2, KIR3DL2, and KIR2DS3. LIM2 is the lens intrinsic membrane 2, a gene that encodes the protein specifically for the eye lens. It is a receptor for calmodulin and functions in cataractogenesis and lens development. A study found a potential relationship between old age and the LIM2 gene, as it is involved in the development of age-related cataracts [22]. This explained the recent findings in POCD patients of an older age group.

Erythroferrone (ERFE) is a protein coded by the gene ERFE. As the primary erythroid regulator of hepcidin, erythroferrone plays a role in iron homeostasis. Hepcidin regulates total body iron content and plasma levels in normal conditions. This ensures iron flows are available whenever there is a need to develop red blood cells while preventing the potential toxicity of excessive extracellular iron. The release of erythropoietin induced the ERFE expression, with erythroblast being the highest ERFE-expressing cell type [23]. During a bypass procedure, iron homeostasis is disturbed by several means, such as blood loss through the incision, blood shunt to a bypass machine, and hemolysis. This hemopoietic stress will cause an iron loss in blood plasma, jeopardizing the most iron-extensive process, erythropoiesis. In the situation of high iron demand, erythroblast produced ERFE, which inhibited the production of hepcidin, which then allowed ferroportin to transport iron into the bloodstream [24]. This ensures iron regulation during surgery is maintained at an optimum level to avoid the risk of multi-organ failure due to hypoxia.

A study proves that preserving blood flow to the brain has lesser adverse outcomes post-open-heart surgery [25]. Maintaining cerebral blood perfusion will ensure a stable blood flow to the brain. This will control a continuous oxygen supply. The meta-analysis found that high oxygen levels contribute to a significantly lower risk of getting POCD [26]. Hence, we hypothesized that massive blood loss during surgery reduced blood flow to the brain, thereby reducing oxygen supply that may cause tissue hypoxia and inflammatory reactions, which led to POCD afterward. In the case of POCD, the demand for erythroferrone is higher to maintain iron homeostasis, as reflected by the expression postoperatively.

KIR2DS2, KIR3DL2, and KIR2DS3 genes are killer cell immunoglobulin-like receptors (KIR), with 2 Ig domains and short cytoplasmic tail 2, 3 Ig domains and long cytoplasmic tail 2, and 2 Ig domains with short cytoplasmic tail 3, respectively. They are protein-coding genes that are classified according to their extracellular immunoglobulin number (either 2D or 3D) and by long (L) or short (S) cytoplasmic domain. KIR are receptors on natural killer (NK) cells together with their ligands, human leukocyte antigen (HLA). NK cells with the KIR2DS2 immunogenotype were found to be efficient killers of glioblastomas, a brain cancer cell [27]. NK cells were known to be involved in multiple neurodegenerative diseases such as Alzheimer’s disease (AD), Parkinson’s disease (PD), and amyotrophic lateral sclerosis (ALS) [28,29,30]. A study found a correlation between the haplotype combination of KIR2DS2/KIR2DL2/HLA-C1 with lower MMSE scores, representing the condition of AD [28]. On the other hand, KIR2DS3 was found to correlate with acute myeloid leukemia (AML), where the frequency was significantly lower in the AML group compared to the control [31]. Besides, KIR3DL2 was recently found to be the biomarker for Sezary syndrome, a potent type of cutaneous T-cell lymphoma. The percentage of KIR3DL2+ cells at diagnosis with >85% correlated with reduced disease-specific survival [32]. In ALS, the knockdown of KIR3DL2 receptors on the HLA-F molecule on human astrocytes enhanced motor neuron death [33].

All KIR genes play an essential role in the regulation of immune responses. The genes were downregulated in our POCD group. Although the link between these downregulations and POCD occurrence was unclear, clinicians anticipated the condition due to the possibility that the intervention during surgery would impact patients’ immune responses. Blood contact with foreign surfaces from the bypass circuit, surgical trauma, endotoxin, and ischemia may provoke systemic inflammatory response syndrome (SIRS) [34]. SIRS during surgery may develop complications such as myocardial dysfunction, neurological and renal dysfunction, respiratory failure, bleeding disorders, and multi-organ failure [35]. Multiple interventions during cardiopulmonary bypass (CPB) may prevent the secretion of pro-inflammatory cytokines and lessen the complement-mediated activation of neutrophils [34]. As a result, immune response regulator genes such as KIRs were downregulated, mirroring the actual clinical intervention performed on selected CABG patients. We hypothesized that these genes may be related to neuroinflammation. Studies showed that NK cells play a role in Parkinson’s disease, where NK cells inhibit microglial transactivation, which later reduces neuroinflammation [36]. It is suggested that NK cells have a neuroprotective effect on PD pathogenesis [27]. We speculate that the downregulation of KIR genes may interfere with the neuroprotective effect of NK cells, hence increasing the risk of POCD.

The limitation of this study includes the time of blood collection that differs for every patient, which might change the gene expression level. Blood collections were set for day three postoperatively, without a specific time. The day for postoperative assessment also differed from patient to patient, which may influence the overall results. Furthermore, the significantly dysregulated genes in our results were not as numerous as in other studies, implying that potential genes of interest may be overlooked. 

## 5. Conclusions

It was found that there were altered gene expressions postoperatively in high-risk POCD patients receiving dexmedetomidine. The current study provides the gene expression profiles obtained from microarray analysis, which can be used to find potential proteins for predictive biomarkers. The dysregulated genes were an actual representation of cardiopulmonary bypass, which is probably involved in the mechanism of POCD. The genes found in this study were associated with POCD development, yet further elucidation is required for a thorough understanding of its importance in clinical settings. Pathway analysis for these genes is suggested to further understand their roles, particularly in the mechanisms and networks involved.

## Figures and Tables

**Figure 1 ijerph-20-01457-f001:**
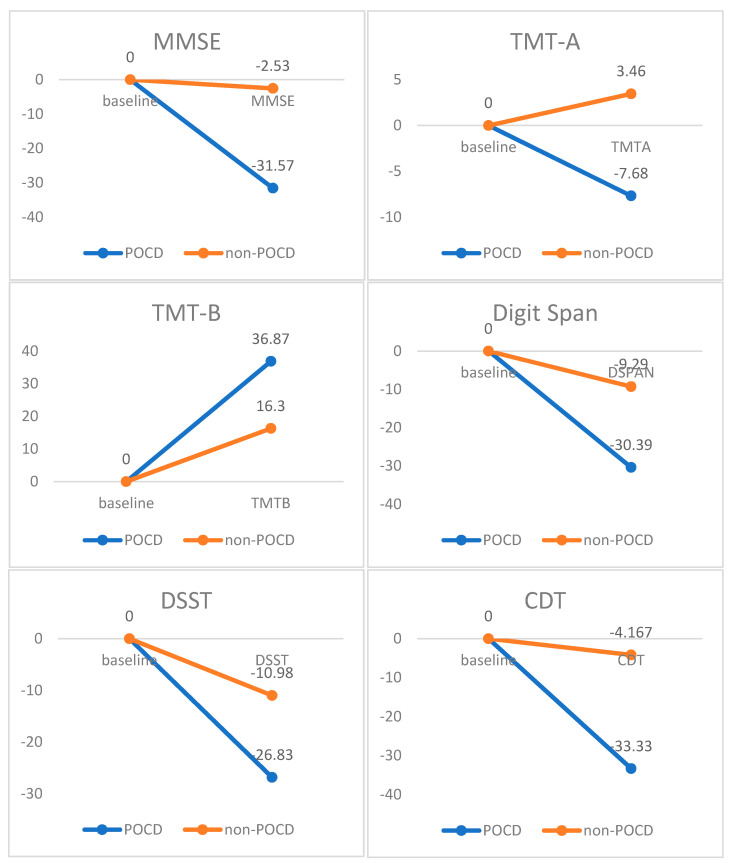
Percentage differences between POCD and non−POCD patients in all tests. MMSE = Mini−Mental State Examination; TMT = Trail Making Test; DSST = digit symbol substitution test; CDT = clock drawing test.

**Figure 2 ijerph-20-01457-f002:**
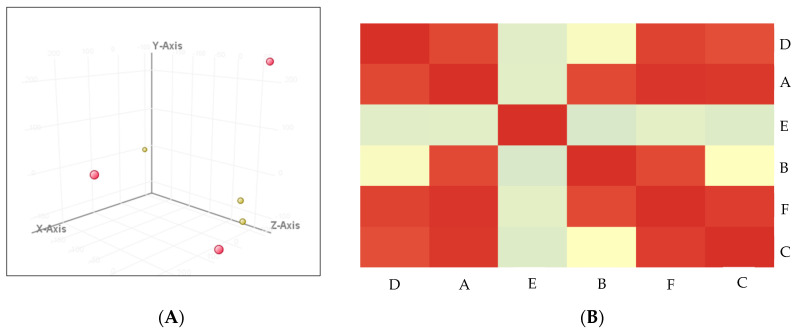
Sample quality. (**A**) PCA plot. Yellow dots are non−POCD samples; red dots are POCD samples. (**B**) Correlation analysis. A, B, and C are non−POCD samples, while D, E, and F are POCD samples. Darker red means higher correlation between samples.

**Figure 3 ijerph-20-01457-f003:**
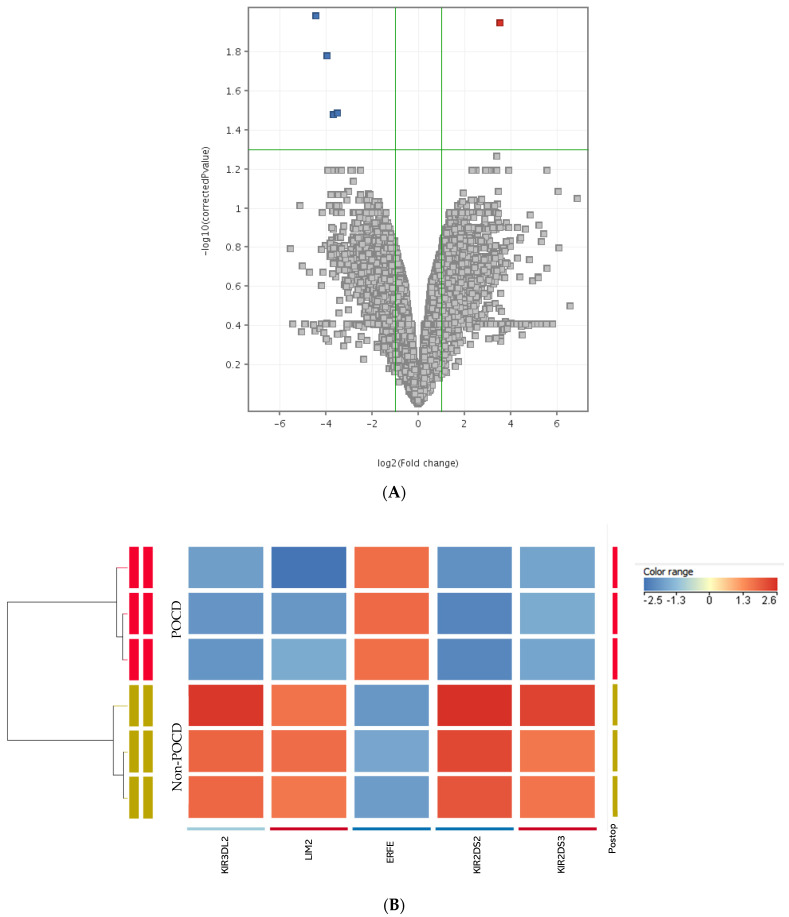
Distribution of differentially expressed genes. (**A**) Volcano plot showing upregulated (red) genes on the top right and downregulated (blue) genes on the top left of the plot. Genes in grey color are not significantly expressed. (**B**) Heatmap showing the upregulated (red) and downregulated (blue) genes in POCD and non−POCD condition.

**Figure 4 ijerph-20-01457-f004:**
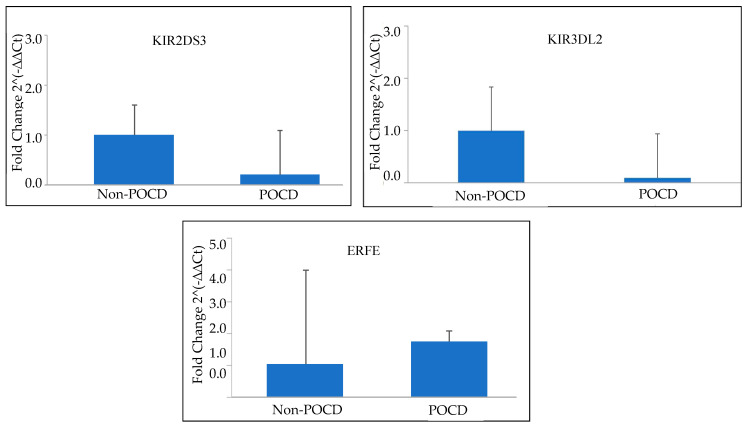
Bar graphs representing expression levels of KIR2DS3, KIR3DL2, and ERFE genes by quantitative real-time RT-PCR in POCD and non-POCD groups define by their fold change.

**Table 1 ijerph-20-01457-t001:** Demographic, intraoperative characteristics, and cognitive assessment scores of patients in the POCD and non-POCD groups.

Item	Non-POCD (*n* = 38), Mean ± SD/*n* (%)	POCD (*n* = 8), Mean ± SD/*n* (%)	*p*-Value
Age (years)	56.21 ± 13.191	66.00 ± 5.976	0.047 *
Gender			
Male	23 (79.3)	6 (20.7)	0.366
Female	15 (88.2)	2 (11.8)	
Education (years)	11.82 ± 2.710	8.00 ± 4.036	0.002 *
Body Mass Index (BMI)	26.44 ± 4.199	28.34 ± 13.07	2.44
Bypass time (minutes)	127.24 ± 72.431	123.86 ± 26.55	0.905
Cross Clamp time (minutes)	99.35 ± 63.141	84.86 ± 28.956	0.559
Preoperative scores			
MMSE	28.34 ± 2.096	25.50 ± 2.563	0.002 *
TMT-A (seconds)	49.16 ± 26.054	69.38 ± 29.306	0.57
TMT-B (seconds)	107.53 ± 56.125	162.50 ± 62.452	0.017 *
Digit Span	13.63 ± 3.183	10.75 ± 3.370	0.026 *
DSST	53.39 ± 16.839	29.75 ± 14.469	0.001 *
CDT	1.92 ± 0.273	1.75 ± 0.707	0.250
Postoperative scores			
MMSE	27.63 ± 2.454	18.00 ± 7.483	<0.001 **
TMT-A (seconds)	47.8 ± 22.512	71.80 ± 47.484	0.064
TMT-B (seconds)	123.31 ± 71.160	184.20 ± 45.069	0.072
Digit Span	12.08 ± 3.631	6.88 ± 4.853	0.001 *
DSST	47.56 ± 19.166	23.14 ± 14.565	0.003 *
CDT	1.86 ± 0.487	1.14 ± 1.069	0.007 *

POCD, postoperative cognitive dysfunction; MMSE, Mini–Mental State Examination; TMT, Trail Making Test; DSST, digit symbol substitution test; CDT, clock drawing test. * *p*-value significant at *p* < 0.05 using independent *t*-test. ** *p*-value significant at *p* < 0.001 using independent *t*-test.

**Table 2 ijerph-20-01457-t002:** Purity, concentration, and integrity of RNA samples.

Samples	Purity	Concentration	RIN Number
A	1.832	43.2	8.80
B	1.855	61.6	6.80
C	1.857	46.8	8.00
D	1.892	25.2	8.80
E	1.842	135.6	8.00
F	1.816	63.2	8.40

**Table 3 ijerph-20-01457-t003:** Differentially expressed genes in the POCD group.

No	Fold Change	*p*-Value	Regulation	Gene Symbol	Description
1	11.606633	0.01127	Up	ERFE	erythroferrone
2	−11.387419	0.03253	Down	KIR2DS2	killer cell immunoglobulin-like receptor, two Ig domains, and short cytoplasmic tail 2
3	−12.647894	0.03299	Down	LIM2	lens intrinsic membrane protein 2
4	−15.381129	0.01660	Down	KIR3DL2	killer cell immunoglobulin-like receptor, three Ig domains, and long cytoplasmic tail 2
5	−21.730328	0.01028	Down	KIR2DS3	killer cell immunoglobulin-like receptor, two Ig domains, and short cytoplasmic tail 3

## Data Availability

All data are available within the article and publicly accessible.

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
