# Peer review of "Microarray Profiling of Differentially Expressed Genes in Coronary Artery Bypass Grafts of High-Risk Patients with Postoperative Cognitive Dysfunctions"

_ijerph, 2023, doi:10.3390/ijerph20021457_

Round 1

Reviewer 1 Report

In this study, the authors investigated the incidence of POCD using specific neurocognitive assessments and aimed to understand its associated key differentially expressed genes using the microarray technique, in high-risk coronary artery bypass graft patients.

The manuscript as a whole is good but needs revision to increase its quality.

·         The introduction should highlight the novelty of the paper. 

·         Results and discussion need elaboration.

·         There are language errors that need editing.

·       Specify your abbreviations along with the whole manuscript (for easier reading).

·         What are your significant values?

·         What is the limitation of this investigation?

Reviewer 2 Report

“Microarray profiling of differentially expressed genes in coronary artery bypass grafts of high-risk patients with postoperative cognitive dysfunctions”   by Yazit et al.

In this manuscript the authors examine the differential expression of certain genes in coronary artery bypass patients exhibiting post-operative cognitive dysfunction in a prospective cohort study. They carry out microarray analysis to profile the differential gene expressions. The authors report here that out of five genes showing differential expression between patients having and not having cognitive dysfunction, one was upregulated while the remaining four were downregulated in patients with POCD.  The authors try to explain the differential up- or downregulation of the genes with respect to the function of the genes and surgical trauma.  They also state that POCD was more prevalent in older and less educated patients.

In the background of not much being known about the pathogenesis and mechanism of POCD, this study does shed some light on which genes may be involved in leading to cognitive dysfunction. It opens up the path for more extensive gene expression profiling in POCD patents that could potentially lead to identification of proteins for POCD diagnostic and predictive biomarkers. This gives the importance to this specific study by the authors.

The manuscript is well written with minimal grammatical errors.  Conclusions are well drawn from the observations and the data is well presented visually. Overall it is a good piece of work.

Certain minor points that need t be corrected are as follows:

1.      Line 27:  The authors write “According to results of the present study, POCD can still occur, but at a lower incidence rate 27 than previously reports.”  The meaning of this sentence is not very clear. What do they mean by POCD can STILL occur?  I would recommend rephrasing of this sentence to make the meaning very clear and obvious.

2.      Lines 135-136:  The word “briefly/brief” has been used twice here, which does not sound very good.  The authors can rephrase the sentences.

3.      Lines 255-256: The authors write “a gene that encodes the protein for the eye-lens specific”.  This sentence appears to be incomplete.

4.      Line 264: “ ERFE expression was expressed faster when erythropoietin (EPO) was released,” This sentence does not sound right with expression/expressed.  Please correct this sentence.

Reviewer 3 Report

Dear authors,

Your study addresses an intriguing and, as you very well mentioned, not enough studied after-effect of massive surgeries: the postoperative cognitive decline. The study is laborious and the results are presented clearly with well-conceived graphs. The subject is difficult and not easy to navigate given the diversity of data on one hand and the lack of a clear direction on the other hand (within that data).

However, there are a couple of issues that I want to highlight and which I believe can improve the overall clarity of the presentation:

In the Introduction

1. line 33  - I believe that a short enumeration of the major clinical manifestations of POCD would be welcome. It is not a common subject and can help understand the choice of tests in the study. 

2. The introduction deals with two research directions present in the literature so far: gene expression and neuroinflammation. Yet, the text jumps from one to the other and then back. I suggest you finish presenting current findings for one and then deal with the next. Also, a little more detail on the expression of genes related to inflammatory response would also help explain and back up your own findings. 

3. The introduction of dexmedetomidine is very inconsistent and doesn't underline the importance it has in your own study. 

In Materials and Methods

1. The inclusion criteria, except age (without explanation), are missing. It has to be opposite to exclusion criteria, which are clearly presented. 

2. Line 97: "predisposed cognitive decline": you probably mean a predisposition to cognitive decline or a baseline poor cognitive performance.

3. The choice of timing for testing is not explained. Why is the 7th day after surgery (medical considerations? psychological considerations? other?). The day prior to the day of surgery seems a difficult day from a psychological point of view and may have an influence on the overall results. PLease motivate the choice of the day. 

4. "The blood samples were collected one day before surgery and three days after surgery." The first one makes sense being the day of preoperative testing, but why is the second sample collected on day 3? At this time you don't know if the patient is in one group or the other (the postoperative cognitive testing takes place on day 7). Is this the reason why you have just 3 POCD samples instead of 8? (the number of patients that showed POCD)

In the Discussion and Conclusion

1. There is no mention of the possible relationship between massive intraoperative blood losses and subsequent oxygen supply to the brain and the POCD. 

2. The limitations of the study should include the small number of samples and the limitations of data collection (as mentioned above). 

3. The discussion should emphasize the finding of neuroinflammation and related gene expression, also found in the study. 
